# Aphid Recognition and Counting Based on an Improved YOLOv5 Algorithm in a Climate Chamber Environment

**DOI:** 10.3390/insects14110839

**Published:** 2023-10-28

**Authors:** Xiaoyin Li, Lixing Wang, Hong Miao, Shanwen Zhang

**Affiliations:** College of Mechanical Engineering, Yangzhou University, Yangzhou 225127, China

**Keywords:** pest recognition, pest counting, YOLOv5, deep learning

## Abstract

**Simple Summary:**

The pepper [*Capsicum annuum* L. (Solanales: Solanaceae)] is one of the most economically important vegetable crops and the demand for peppers has been increasing. In the process of pepper crop growth, the types and quantity of insect pests are increasing, which has become the first natural biological disaster threatening crop yield and quality. In particular, the green peach aphid [*Myzus persicae* Sulzer (Hemiptera: Aphididae)] is one of the most threatening insect pests in pepper cultivation, which causes a lot of damage, such as necrosis, wilting, chlorosis, and defoliation. Hence, accurately recognizing and counting the aphids is essential for ensuring the excellent productivity of pepper crops. Due to the time-consuming and inefficient work of traditional methods, there have been deep learning methods for crop pest recognition and counting in recent years. The YOLO algorithm is one of the most effective deep learning algorithms used for object detection. To investigate aphid recognition and counting of pepper crop growth, this paper develops an improved YOLOv5 approach based on a deep convolutional neural network (CNN) in the climate chamber environment. Experimental results in the aphid dataset indicate that the proposed method can achieve significant improvements in terms of recognition accuracy and counting performance.

**Abstract:**

Due to changes in light intensity, varying degrees of aphid aggregation, and small scales in the climate chamber environment, accurately identifying and counting aphids remains a challenge. In this paper, an improved YOLOv5 aphid detection model based on CNN is proposed to address aphid recognition and counting. First, to reduce the overfitting problem of insufficient data, the proposed YOLOv5 model uses an image enhancement method combining Mosaic and GridMask to expand the aphid dataset. Second, a convolutional block attention mechanism (CBAM) is proposed in the backbone layer to improve the recognition accuracy of aphid small targets. Subsequently, the feature fusion method of bi-directional feature pyramid network (BiFPN) is employed to enhance the YOLOv5 neck, further improving the recognition accuracy and speed of aphids; in addition, a Transformer structure is introduced in front of the detection head to investigate the impact of aphid aggregation and light intensity on recognition accuracy. Experiments have shown that, through the fusion of the proposed methods, the model recognition accuracy and recall rate can reach 99.1%, the value *mAP*@0.5 can reach 99.3%, and the inference time can reach 9.4 ms, which is significantly better than other YOLO series networks. Moreover, it has strong robustness in actual recognition tasks and can provide a reference for pest prevention and control in climate chambers.

## 1. Introduction

The pepper [*Capsicum annuum* L. (Solanales: Solanaceae)] is one of the most economically important vegetable crops, and it is widely cultivated worldwide, including in China, the United States, Spain, and Mexico [1,2]. With global population growth, the demand for peppers has been increasing. To increase the yield and improve the quality of the pepper it is interesting to investigate it in a climate chamber. Due to the humid environment of the climate chamber it is highly susceptible to insect pests, which is a hindrance to pepper cultivation in the climate chamber [3]. Hence, attention has been drawn to the use of artificial climate chambers for pepper pest control. The green peach aphid [*Myzus persicae* Sulzer (Hemiptera: Aphididae)] is one of the most threatening insect pests in pepper cultivation, which causes a lot of damage [1,2,4]. Further, the most serious damage is done by a large number of viruses that the aphid may transmit. The migration of aphids makes them densely distributed on the back of leaves or curled up in young new leaves and hidden in various positions of pepper leaf. They have strong aggregation, making it difficult to count them and seriously affecting indoor pest investigations in the climate. Therefore, there is a pressing need to develop a precise and convenient approach for recognizing and counting pepper leaf aphids.

To tackle the issue, various approaches have been used to recognize and count insect pests. Insect detection can work both in terms of the detection of the insects or the actual damage they cause. The first and most direct method is artificial pest recognition and counting, which relies on the human eye for judgment. Usually, it is performed by agricultural experts. Due to the density of pests, the implementation workload is large and prone to misjudgment. Other traditional methods, such as the carbon dioxide measurement method [5], passive acoustic detection [6], ultra-wideband detection [7], and X-ray computed tomography [8], have been successfully used to detect insect pests. These methods are usually integrated into the detection systems and appropriate equipment to support them must be developed. For instance, a method for trapping and detection using an automated pheromone attractant lure with a battery-powered SightTrap™ system and its accompanying software ForesightIPM™ has been designed to detect the presence of *Tineola bisselliella*. However, there was a long development cycle and high cost for this detection equipment and system. Additionally, the detection accuracy and time may be improved.

The second method uses traditional machine learning-based insect pest monitoring, which is mainly based on computer vision [9,10,11,12,13,14,15,16]. These algorithms are mostly based on pest features combined with classifiers for screening, which cannot be separated from large amounts of parameter tuning and image processing work. They require extracting many image features of insect pests, which makes the detection process more complicated. Moreover, there are still limitations, such as robustness performance, computing efficiency, inaccuracy, and lack of generalization capability in complex environments [9,10]. In addition, due to the randomness of pest migration (like the green peach aphid), simply using insect traps to achieve pest counting cannot accurately reflect the internal pest problems in the environment [17].

With the advancement of Artificial Intelligence, the third method uses deep learning-based pest identification and counting, in which pest detection tasks can be unified into an end-to-end feature extraction approach. At present, the commonly used deep learning pest detection networks are CNNs (convolutional neural networks) [18,19,20], Mask R-CNN (mask region with the convolutional neural network) [21], SSD (Single shot multi-box detector) [22], and the YOLO algorithm (You only look once) [23]. In particular, object detection algorithms mainly based on YOLO series have been successfully applied in the agricultural pest identification field. The YOLO algorithm is one of the object detection techniques which has faster calculation speed and higher recognition accuracy [24]. The first YOLO version (YOLO v1) was proposed by Redmon et al. for computer vision and pattern recognition [25]. Then, YOLOv2 was used to detect and count pest insects in a pest monitoring system for agriculture; it had an average accuracy of 52.5% and a computation time of 8-10 min per image [26]. To solve tomato pest detection in practical scenarios, a YOLOv3 network was presented, and a recognition accuracy of 92.39% was reached [27]. Chen et al. employed a YOLOv4 model to develop an automatic system of pest detection and counting and obtained a mean average precision (*mAP*) of 97.55% [28]. Li et al. addressed an improved YOLOv4 model for detecting and identifying the disease and pest lesions on the rice canopy and developed an intelligent monitoring system [29]. Zhang et al. proposed an improved YOLOv5 network for unopened cotton boll detection in the field. The experimental results showed that it can effectively help farmers take effective action in time, reduce crop losses, and therefore increase production [30]. Guo et al. focused on an automatic monitoring scheme to detect vegetable insect pests and developed a YOLO-SIP detector to identify pests in captured images. The result showed that the average accuracy was 84.22% [31]. In addition, Wen et al. proposed an improved YOLOv4 network for multi-category dense and tiny pests, and obtained accuracies of 69.59% for *mAP* and 77.71% for mRecall in a 24-class pest dataset [32]. Mamdouh and Khattab introduced a YOLO-based deep learning framework for detecting and counting the number of olive fruit flies, which outperformed the existing pest detection method [33]. Thus, it can be seen that the YOLO series has shown excellent development prospects and has significant potential in the field of pest detection. To our knowledge, there have been few limited reports of aphid recognition using the YOLO algorithm. For example, Lacotte et al. evaluated the performance of YOLOv4 using precision and recall criteria, and applied YOLOv4 for the detection of aphids in the pesticide-free robotic control process [34]. Xu et al. proposed a lightweight SSV2-YOLO model based on YOLOv5s for the detection of sugarcane aphids in unstructured natural environments and the method reduced the model size [35]. This provides a reference for us to recognize and count the green peach aphid on the peppers in the climate chamber by employing the YOLO method.

Although there are many methods for the detection and counting of many insect pests, it is still challenging to recognize and count aphids in pepper cultivation fields. First, for aphid recognition in a climate chamber environment, due to the small target size and strong aggregation of aphids, when multiple other factors are interfering with crops or a sticky insect board, such as other light-sensitive pests and impurities, this redundant background information is learned along with the network, which seriously affecting the model’s ability to extract aphid features. In addition, the quality of the collected images is easily affected by the lighting in the climate chamber, making aphid recognition and counting challenging. Finally, there exists large intra-class similarity and small inter-class variance in symptom phenotypes, thus it is difficult to recognize and count the aphids.

In response to the above problems, the question of how to combine the YOLO model with actual recognition tasks is an urgent technical challenge that needs to be solved. To achieve aphid recognition and counting in the climate chamber, this paper proposes YOLOv5 as the basic network structure and makes improvements on this basis. The specific improvements are as follows.

(1)To distinguish the characteristic information of aphids from other pests or impurities, a convolutional block attention mechanism (CBAM) is introduced in the YOLOv5 backbone layer. It can enhance the model’s feature extraction ability for targets, making it more focused on aphid target information.(2)To further improve the recognition performance of the model for different degrees of aphid aggregation, a Transformer structure is introduced in front of each detection head. This can improve the recognition ability of the model for small aphid targets under different light sensitivity and aggregation levels.(3)To highlight the overall detection accuracy of the model in different aphid detection scenarios (aphid recognition counting on leaves and recognition counting on lure boards), the idea of a bi-directional feature pyramid network (BiFPN) is employed. Changing the neck PANet to a BiFPN structure reduces the loss of aphid feature information caused by the fusion of contextual information, thereby improving model accuracy.

The remainder sections of this paper are organized as follows: Materials and methods of aphid recognition and counting are addressed in Section 2. An improved YOLOv5 architecture for the issue is proposed in this section. Experimental results and analysis are presented to verify the feasibility and efficiency of the proposed approach in Section 3. The conclusion is presented in Section 4. For convenience, the main abbreviations in this paper are summarized in Table 1.

## 2. Materials and Methods

### 2.1. Aphid Dataset

The green peach aphids [*Myzus persicae* Sulzer (Hemiptera: Aphididae)] came from the laboratory of the School of Horticulture and Plant Protection, Yangzhou University, and were propagated on the peppers for more than 3 generations in the insect-rearing room. By using an RGB camera with a high resolution provided by China Daheng (Group) Co., Ltd., in Beijing, China, we captured 1000 original pepper leaf aphid images at different growth stages of the green peach aphids. The temperature, humidity, and carbon dioxide required by the pepper in the natural environment are simulated through a temperature, humidity, and ventilation system. The integrated LED light group is used to supplement the required light intensity for the pepper. The temperature control range is 10–40 °C, the humidity control range is 60–90% RH, and the lighting intensity is 0–40,000 Lux. The actual view of the climate chamber is shown in Figure 1.

Two aphid image acquisition methods were considered based on the different growth stages of peppers grown in a climate chamber. One is the collection of images of aphids on pepper crops and the other is the collection of images of aphids on insect traps.

(1) Collection of images of aphids on crops: In the early stage of aphid outbreaks the degree of aphid aggregation is relatively low, and only a small number of aphids appear on the leaves of peppers. The aphids were captured by taking photos of each pepper leaf, as shown in Figure 2. The images included two aggregation types, (a) mild aggregation and (b) moderate aggregation, which were taken under two kinds of light intensities, respectively.

(2) Collection of images of aphids on insect traps: In the late stage of the aphid outbreak the aphids multiplied rapidly, and the aphids began to spread rapidly to the roots, fruits, and other parts of the pepper. The aphids propagate continuously and accumulate seriously, which makes it difficult to count them. The yellow trap board was arranged under the pepper crop, which made the aphids fall on the trap board. The camera was used to collect the aphid information from the trap board. The insect trap plate specification was set to 15 × 10 cm. Due to different lighting requirements during different stages of pepper growth, different degrees of photosensitive imaging are generated. As aphids gather, they exhibit different degrees of aggregation distribution, which can be seen in Figure 3. This corresponded to the distribution of aphids under mild aggregation, severe aggregation, and high aggregation. Figure 3a represents an image that was captured under high lighting conditions below 30,000 Lux, Figure 3b represents an image that was captured under medium lighting intensity below 15,000 Lux, and Figure 3c represents an image that was captured under low light intensity below 5000 Lux.

When the data sample is limited for the aphid dataset, it is necessary to expand the data set to avoid the overfitting of the model. There are various dataset enhancement methods, such as Mosaic and GridMask. Mosaic data augmentation concatenates images through random scaling, cropping, and distribution [36]. GridMask is an information removal image enhancement method that randomly and evenly removes partial areas of information [37]. In our paper, we combined Mosaic and GridMask to enhance the dataset, which is shown in Figure 4. By combining Mosaic and GridMask with spatial transformation and information removal, data augmentation can further enrich the dataset. Using the above image enhancement method, the dataset was expanded to 1500 images, and the training, validation, and testing sets were divided according to the ratio of 8:1:1. In addition, agricultural experts labeled the collected images of aphids by using the labeling tool in [38]. The annotations of images were saved in the format of .xml.

### 2.2. Methods

#### 2.2.1. Original YOLOv5 Model

YOLOv5 differs from the RCNN [28] series algorithm; as a single-stage target recognition algorithm, it is easy to configure the environment and train the dataset. Due to the flexible control of the model size, it can achieve both high recognition speed and accuracy [24]. According to the number of parameters, YOLOv5 can be divided into five types: YOLOv5n, YOLOv5s, YOLOv5m, YOLOv5l, and YOLOv5x. Since the YOLOv5s model has the smallest parameters and is easier to deploy and train, we used YOLOv5s in the study. The original structure is shown in Figure 5. It consists of the input, backbone, neck, and output layers. In the input layer, aphid images need to be data enhanced for training; in the backbone layer there are the Focus structure, Conv module, C3 module, and SPP module. The Focus structure achieves downsampling between adjacent pixels by slicing the input image, thereby reducing computational complexity, and increasing network fitting speed. The Conv module represents the combination of the convolution layer, batch normalization layer, and SiLU activation function. As the main module for residual feature learning in the YOLOv5s model structure, the C3 module enhances the network’s feature extraction ability by concatenating features from different channels. In addition, the Bottleneck module plays an important role in the process of aphid feature extraction. A schematic diagram of the Bottleneck module structure is shown in Figure 5, which integrates 1 × 1 convolution and 3 × 3 convolution with the input for feature fusion. It can reduce the number of parameters and computation in the model. The path aggregation network (PANet) is one common structure in the neck layer. The neck is designed to make better use of the features that the backbone extracts at different stages. In the output layer, the feature map after processing is predicted.

#### 2.2.2. Proposed YOLOv5 Architecture

In this section, the YOLOv5s model was improved to achieve recognition and counting of aphids under different light sensitivity and aggregation levels. The proposed model is shown in Figure 6. By introducing the CBAM attention mechanism after the C3 module in the backbone layer, the interference of redundant information and aphid recognition is reduced. By adding a Transformer module (represented by TR in the figure) in front of each detection head, global information can be better obtained to solve the problem of difficulty in identifying and counting aphids due to strong light and varying degrees of aggregation. Using the BiFPN neck structure instead of the original PANet neck, the fusion of contextual information further improves the accuracy of aphid recognition.

(1)CBAM attention mechanism:

CBAM is a lightweight attention mechanism that combines channel and space, consisting of a Channel Attention Module (CAM) and a Spatial Attention Module (SAM) [39]. By multiplying the channel features and spatial features of the C3 module, the spatial and channel dimensions can be compressed while the channel and spatial dimensions remain unchanged, allowing aphid information to be primarily focused on learning. The process of aphid feature extraction combined with the attention mechanism is shown in Figure 7. In the first step, the aphid channel information is extracted through CAM. For the feature map (H × W × C) generated after C3, the aphid information with 1 × 1 × C can be obtained by using an MLP (Multilayer Perceptron) based on parallel calculations of max pooling and average pooling, then the corresponding feature mapping is achieved. Through sigmoid activation (σ) and weighting operation (+), the channel attention feature *Fc* is generated; after adaptive correction by multiplying *F*, the aphid channel feature output $F^{\prime }$ is produced. The calculation formulae are shown in Equations (1) and (2), where ⊗ represents the multiplication operation between matrices.
(1)Fc=σ{MLP[Maxpooling(F)]+MLP[Avgpooling(F)]}
(2)$$F^{\prime }=F⊗F_{c}$$

Based on obtaining the output of aphid channel features, convolutional layers are used to extract the features concatenated by Maxpooling and Avgpooling. According to the operation procedure of SAM, the aphid spatial attention feature *F_s_* can be obtained through sigmoid activation (σ). The calculation formula is as follows.
(3)$$F_{s}=\sigma \{\mathrm{conv}[\mathrm{Maxpooling}(F);\mathrm{Avgpooling}(F)]\}$$

Finally, based on the CAM and SAM operations mentioned above, the refined aphid channel spatial feature output $F^{\prime \prime }$ is obtained by multiplying it with the aphid channel feature output $F^{\prime }$ again. The calculation formula is as follows.
(4)$$F^{\prime \prime }=F^{\prime }⊗F_{s}$$

(2)Transformer module:

The structure of the Transformer encoder and decoder has achieved good results in natural processing languages [40]. Compared with ordinary convolution, the linear projection method has less computational complexity and can better extract global information about the target, making the Transformer encoder widely used in the field of target detection. By adding a Transformer encoding module in front of each detection head, the extraction and expression of global features of the target can be achieved. In addition, the appropriate contextual information is introduced to supplement the aphid information. The Transformer encoder can achieve the accurate prediction of aphid position and category information with different aggregation levels in complex lighting environments. Additionally, it can reduce false detection and missed detection. The Transformer encoder is shown in Figure 8. It includes two main parts, namely, the multi-head self-attention mechanism (MSA) and multilayer perceptron (MLP). The MSA consists of multiple groups of attention. A single attention mechanism *Z_n_* (*n* = 1, 2, 3…) updates and concatenates the query values (*Q*), key values (*K*), and weight values (*V*), containing global feature information from different subspaces. After performing linear regression, an MSA is formed by concatenating and fusing the subset features of each attention mechanism. The MSA can overcome the impact of complex information such as lighting and aggregation on aphid recognition accuracy. The calculation formula is as follows:(5)$$Z_{n}=\mathrm{Attention}(Q,K,V)=\mathrm{softmax}(\frac{QK^{T}}{\sqrt{d_{k}}})V$$
(6)$$\mathrm{MSA}(Q,K,V)=\mathrm{Concat}(Z_{1},Z_{2},\ldots ,Z_{n})W^{0}$$
where *W*^0^ is the weight of the updated parameter, *QK^T^* is used to calculate the correlation degree between the image set composed of the input image feature set and other feature sets, and *d_k_* is the input dimension.

Finally, the non-linear output of features is performed through MLP to enhance the expression of the self-attention mechanism, capturing aphid contextual information, and reducing global information loss. The normalization layer standardizes the input to accelerate model convergence speed.

(3)BiFPN structure:

In the section, a bi-directional feature pyramid network (BiFPN) is considered in the proposed model. Different from the feature fusion mode of PANet adopted by YOLOv5, BiFPN assigns different weights to different layers based on the aggregation of PANet’s top-down and bottom-up paths [41]. It removes some nodes with low contribution rates to the features and adds lateral jump connections between the same scales, simplifying the network while achieving high-level feature fusion. Thus, it makes the model interconnected with aphid recognition tasks on pepper crops and insect traps, which improves the computation efficiency and detection accuracy of the neural network model in different recognition scenarios. Structural diagrams of PANet and BiFPN networks are depicted in Figure 9. For aphid features with different resolution information on pepper crops and insect traps, BiFPN uses fast normalized fusion to allocate different weights for different feature inputs, as shown in Equation (7). Among them, $I_{i}$ represents the input feature matrix of the previous node, $W_{i}$ represents the corresponding weight parameter, and *j* represents the input weight parameter of this node; ∈ was set to 0.0001.
(7)$$O=\sum_{i}\frac{W_{i}}{\in +\sum jW_{i}}\cdot I_{i}$$

On this basis, the set of bidirectional cross-scale connections learns more aphid feature details. The calculation formula for intermediate node feature output $p_{2}^{out}$ is as follows, where $p_{1}^{out},p_{2}^{out},\mathrm{and}\;p_{3}^{out}$ correspond to the output of nodes 1, 2, and 3, respectively. $p_{1}^{in},p_{2}^{in},\mathrm{and}\;p_{3}^{in}$ correspond to the input of nodes 1, 2, and 3, respectively. $p_{2}^{td}$ represents the feature output of the second intermediate layer, Conv represents convolution operation, and $w_{1},w_{2},\;\mathrm{and}\;w_{3}$ represent the feature weights from different layers.
(8)p2td=Conv[w1×p2in+w2×Resize(p3in)w1+w2+∈]
(9)p2out=Conv[w1×p2in+w2×p2td+w3×Resize(p1out)w1+w2+w3+∈]

### 2.3. Experimental Settings

The experiments were executed using the Windows 10 operating system, with processor information from Intel (R) Xeon (R) Silver 4116 CPU @ 2.10GHz, graphics card information from NVIDIA Quadro P600, and PyTorch 1.9.0 as the deep learning framework. The Python version used is 3.8.10. The hyperparameter design was as follows: To improve the training speed, the initial learning rate was set to 0.01; momentum and weight-decay coefficients are two important parameters in the optimizer. The former controls the stability of the optimization direction; the larger its value, the more stable the optimization direction. The latter controls the size of L2 regularization, and the smaller its value, the less the contribution of the regulation to the model loss. Thus, we selected the default values of momentum (i.e., 0.937) and weight-decay (i.e., 0.0005); to avoid overfitting, we reduced epochs to 200 iteration times.

### 2.4. Evaluating Indicators

To investigate the effectiveness of the model in aphid recognition and counting, precision (*P*), recall (*R*), average precision (*AP*), and mean average accuracy(*mAP*) were used as evaluation indicators of algorithm reliability and performance, the mathematical expression is as follows:(10)$$P=\frac{TP}{TP+FP}$$
(11)$$R=\frac{TP}{TP+FN}$$
(12)$$AP=\frac{TP+TN}{TP+FN+FP}$$
(13)$$mAP=\frac{1}{Q}\sum_{q=Q}AP(q)$$
where, *TP* represents the number of positive samples predicted as positive samples, *FP* represents the number of negative samples predicted as positive samples, *FN* represents the number of positive samples predicted as negative samples, and *AP* represents the average of all accuracy obtained under all possible values of recall rates. The *AP* value is averaged across all categories when the *IOU* threshold is set to 0.5.

*P* reflects how many real instances are in the positive case results, and *R* reflects how many real instances are recalled. Due to the potential contradiction between *P* and *R*, the performance of the model cannot be evaluated solely based on them. By introducing *AP*, which is the area enclosed by the *P–R* curve and the coordinate axis, the performance of the model is evaluated more accurately. When the *IOU* threshold is set to 0.5, the higher the value *mAP*@0.5, the better the performance of the model.

## 3. Experimental Results and Analyses

### 3.1. Experimental Results of the Performance Indicators

To evaluate the performance of the proposed model, it was compared with YOLOv5 and other YOLO series algorithms in the same environmental configuration. The experimental results are shown in Table 2. The improved YOLOv5 model has an accuracy, recall, and mAP of over 99%, which is significantly improved compared to other models in the YOLO series. Although the inference time has increased compared to the YOLOv5 model, the accuracy, recall, and *mAP* of the improved model have increased by 0.134, 0.096, and 0.12 percentage points, respectively, which is acceptable for actual aphid detection.

To further examine the proposed model, we investigated the P–R curve. The P–R curve is also known as the precision–recall curve, with the *x*-axis representing precision and the *y*-axis representing recall. The more convex the P–R curve is, and the larger the area enclosed by the two axes, the better the performance of the model. It combines excellent precision and recall, and can comprehensively demonstrate the performance of the model. The P–R curves of different models are shown in Figure 10. As shown in the figure, the P–R curve of the improved model is located in the upper right corner of the image, surrounding other models, indicating that the improved model has achieved good performance in aphid detection.

### 3.2. Ablation Experiment Analysis

To further evaluate the impact of the method on the model, ablation experiments were set up for different improvement points to examine the performance of the model, as shown in Table 3. In Table 3, the symbol + represents the module that is considered to be YOLOv5. YOLOv5s was selected as the network architecture. A CBAM, BiFPN, and Transformer were investigated in our model; YOLOv5-A, YOLOv5-B, and YOLOv5-C considered CBAM, BiFPN, and Transformer, respectively. According to Table 3, compared to the original YOLOv5 model, the introduction of a CBAM in the backbone layer had the greatest impact on the accuracy of the model, increasing by 8 percentage points. Based on YOLOv5, the Transformer structure introduced in front of each detection head had the greatest impact on the model’s recall rate and *mAP* value, increasing by 5.6 and 4.9 percentage points. The overall performance of the model has been improved by introducing improved methods such as a CBAM, a BiFPN, and a Transformer for aphid recognition tasks in specific climate chamber scenarios.

### 3.3. Experimental Results of Aphid Recognition and Counting

To test the detection performance of the model in actual environments, experiments were conducted on the performance of the model at three levels: aphid recognition counting on pepper crops, aphid recognition counting on yellow trap boards, and aphid recognition counting under low light conditions on the test set. Due to the small target size of aphids, the experimental comparison effect was presented more intuitively by hiding the label and confidence information of the anchor box. The experimental results are shown in Figure 11, Figure 12 and Figure 13.

(1) Identification and counting of aphids on pepper crops: Through the recognition and counting effects of different models on aphids on the crops shown in Figure 11, we can observe that the improved model obtains the maximum counting number of aphids on crops, i.e., the number of aphids is 27. Furthermore, it can be found that the improved YOLOv5 model can accurately distinguish the shed cortex of aphids from other pests, and effectively distinguish aphids in areas where aphids gather. However, other models in the YOLO series have issues of missed and false detections and lack sensitivity to aphid information with too small targets.

(2) Identification and counting of aphids on yellow trap boards: As shown in Figure 12, the improved model shows good adaptability to small targets on the yellow trap board. Compared to the YOLO series, the counting performance indicator of the proposed model improves by 10.71–34.78%. On the insect trap board, due to the corresponding experimental processing, the target is singular and the background complexity is smaller, making it easy for the model to achieve aphid recognition and counting.

(3) Aphid recognition and counting under low light conditions: As shown in Figure 13, the improved model has a significant effect on small target recognition and can fully recognize aphids in most positions. Under low light intensity, as the brightness of the images decreases, some meridian areas of pepper crops are almost similar in color to aphids, making it difficult to detect some smaller targets. In the proposed model, the attention mechanism is introduced to be beneficial in improving the performance of Aphid recognition and counting.

### 3.4. Discussion

In this study, we proposed an improved YOLOv5 approach based on deep learning to recognize and count the green peach aphids on pepper crops in a climate chamber environment. Our findings demonstrated that an improvement in the overall performance of the proposed model was achieved. Concretely speaking, by introducing a CBAM, the spatial and channel dimensions of aphid features are compressed, and the network model can better obtain the key information of the feature map. Thus, it can effectively improve the accuracy of aphid recognition on pepper leaves. Using the feature fusion method of the BiFPN structure enhances the YOLOv5 neck, further improving the computation efficiency and recognition accuracy for aphids. In addition, by adding the Transformer encoder, the extraction and expression of the global features of the aphid can be better achieved. Additionally, it has less computational complexity. Compared to other YOLO algorithms (such as YOLOv3, YOLOv4, and YOLOv5), our method achieved recognition accuracy and recall rates of 99.1% and 99.1%, a mAP@0.5 of 99.3%, and an inference time of 9.4 ms, which is superior.

By investigating the extent of aphid infestations on pepper crops based on the proposed model, we found that aphid outbreaks and identification environments have an impact on the improved YOLOv5 model in the climate chamber. In the early stage of aphid outbreaks, there were limited aphids on pepper crops. Thus, the method can identify and count aphids on pepper leaves very well by comparing with other YOLO algorithms. During the later stage of the aphid outbreaks, we needed to add yellow trap boards to recognize and count them as the aphids propagated continuously and accumulated seriously. On the yellow trap board, the aphids were singular and the background complexity was smaller than on the pepper crops, making it easier for the model to achieve aphid recognition and counting. In addition, the low light intensity made it difficult to count aphids on pepper crops. Fortunately, due to the introduction of the CBAM, the model can still show perfect performance with regard to aphid recognition and counting.

Additionally, compared to conventional methods such as artificial pest recognition and counting, our aphid recognition and counting model provides an efficient, fast, and cost-effective method for growers. At the same time, it is convenient and feasible for farmers who may not have the expertise to recognize and count the green peach aphids on pepper crops. Finally, it can perform both offline and online detection in the field by being non-destructive.

Through the above improvements to the YOLOv5 algorithm, identification and counting of small aphid targets in the simple climate chamber can be achieved. Despite the favorable performance demonstrated in the task of pepper leaf aphid recognition and counting, there are certain limitations for further research and improvement. When there is large scale pepper cultivation in the field or greenhouse, a mobile platform to mount the recognition device is needed to detect each pepper crop. Unlike in the small chamber environment, the aphids will migrate and assemble on many other plants when there is an outbreak of aphids in the production environment. Due to the spatial distribution of aphids, it is difficult to count aphids. To prevent duplicate counting, the yellow trap boards can be be deployed according to the pepper planting scale. In addition, aphids may be found on back of leaves, or curled up in young new leaves, and hidden in various positions in the open field environment, which will reduce the effectiveness of the YOLO method.

## 4. Conclusions

This paper takes the YOLOv5 network as the core network for green peach aphid recognition. In response to the impact of lighting, aphid aggregation, and their characteristics on aphid recognition in climate chamber scenarios, a CBAM, BiFPN, and Transformer are introduced in the backbone layer, neck layer, and detection layer. The proposed model can improve the original YOLOv5 model’s low accuracy and poor robustness against aphids. Experiments have shown that the accuracy and recall rate of the model reached 99.1% on the small dataset. Compared to other YOLO algorithms, the actual results of model testing show that the improved model can be used effectively for aphid recognition under low lighting conditions, high aphid aggregation levels, and small targets. The proposed method has a certain guiding significance for agricultural pest investigation.

At the same time, we should further take into account the aphid detection efficiency of the proposed method in real-world greenhouse and open-field production environments. On one hand, due to the large production scale, a mobile platform on which the improved YOLOv5 network can be deployed will be developed to detect pepper leaves quickly. It will have a large amount of computation and memory consumption, and the lightweight design of the model will be investigated without changing the recognition accuracy and speed for pest insects. On the other hand, there seems to be an even more difficult problem. Early in the pepper cultivation we learnt that aphids can be found back of leaves, or curled up in young new leaves, and hidden in various positions in the open field environment. To deal with this more difficult problem, several aspects could be taken into account in future next work. First, a multi-lens and multi-angle camera can be employed to expand the scope of visual images; second, a multi-sensor collaboration method could be used to obtain more information features. By identifying the number of aphids and associating them with other environmental factors, a multi-source information fusion model of pest environmental factor disease can be established to monitor the development trend of insect pests. Third, collecting large-scale datasets will mean that the missing information hidden in various positions will be identified by using our proposed YOLOv5 network architecture and the creation of a complete image will be simulated by using artificial intelligence.

In addition, it was noticed that the method application is significant for farmers in the real world. To make the method practically applicable to farmers, an application (APP) capable of deploying the proposed model on various types of mobile devices is needed. Then, based on the results of green peach aphid recognition and counting, decision-making strategies can be generated to guide pesticide spraying. Based on environmental control, the occurrence of pests can be controlled to reduce dependence on pesticides, which is the focus of our next study.

## Figures and Tables

**Figure 1 insects-14-00839-f001:**
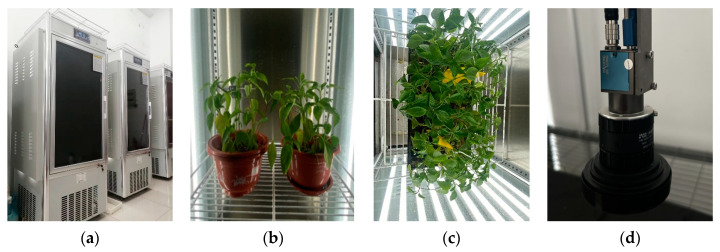
Climate chamber incubation environment diagram. The light, temperature, humidity, and carbon dioxide required by the pepper can be adjusted. Potted pepper and plug-seeding pepper are observed in the climate chamber and their images can be captured by an RGB camera. (**a**) Climate chamber, (**b**) potted pepper, (**c**) plug-seeding pepper, (**d**) RGB camera with high-resolution.

**Figure 2 insects-14-00839-f002:**
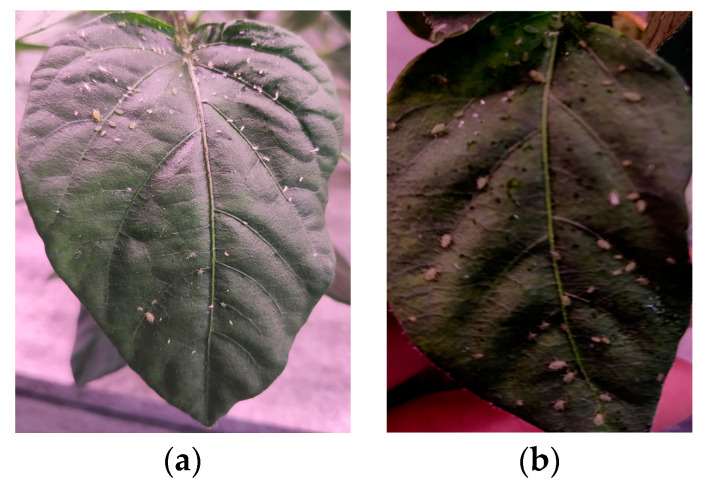
Image of aphids on crops. (**a**) mild aggregation, and (**b**) moderate aggregation.

**Figure 3 insects-14-00839-f003:**
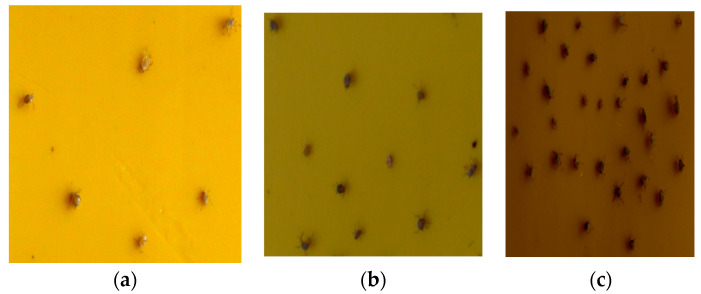
Distribution of different aphid aggregation on the insect board. (**a**) Mild aggregation, (**b**) moderate aggregation, (**c**) high aggregation.

**Figure 4 insects-14-00839-f004:**
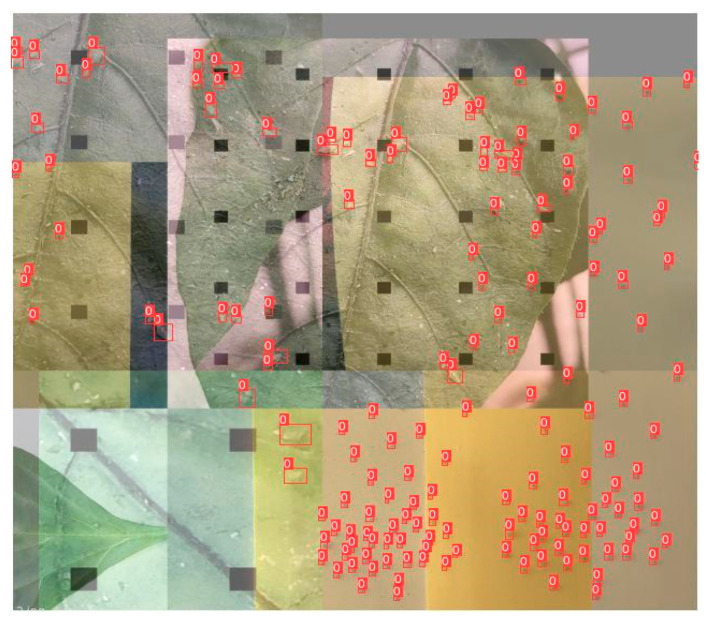
Data enhancement method combining Mosaic and GridMask. Mosaic data augmentation concatenates images and GridMask randomly and evenly removes partial areas of information.

**Figure 5 insects-14-00839-f005:**
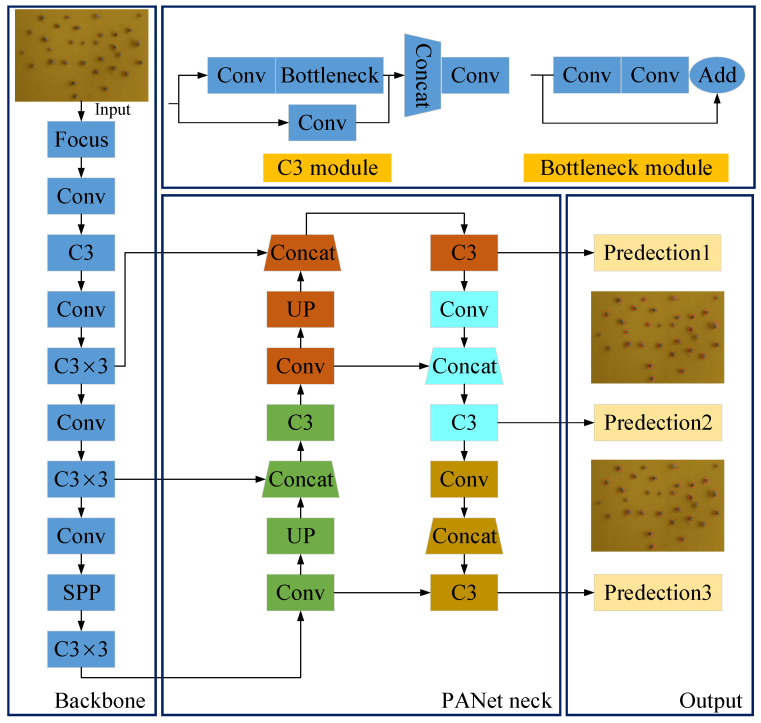
The original YOLOv5s model architecture. It includes the input, backbone, C3 module, Bottleneck module, PANet neck, and output layers.

**Figure 6 insects-14-00839-f006:**
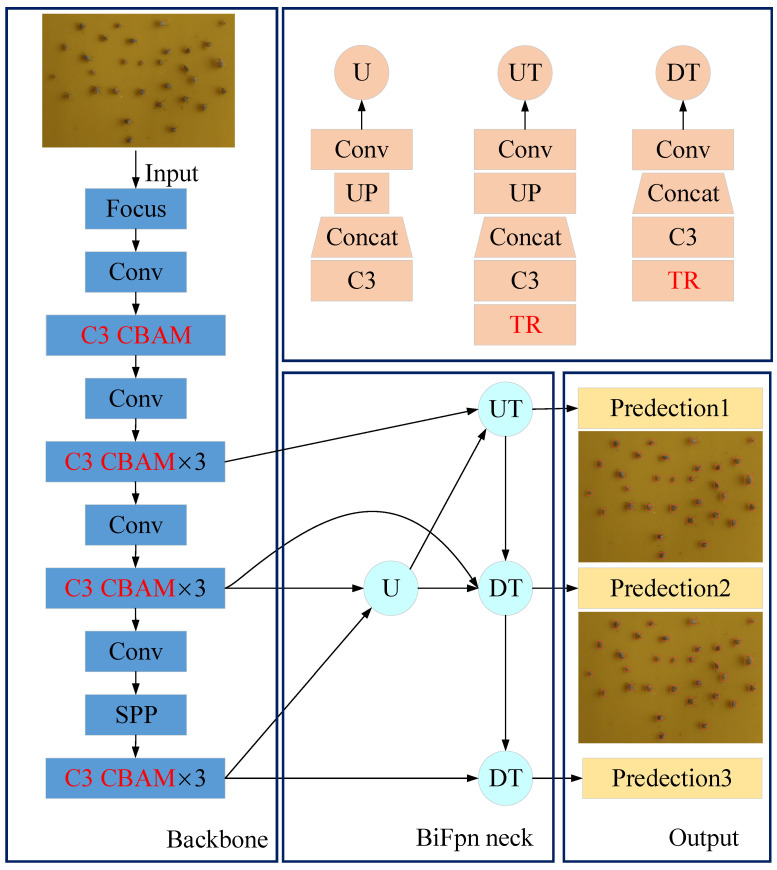
Improved YOLOv5 algorithm structure. The CBAM attention mechanism is introduced to the original YOLOv5 architecture, and the Transformer module is added to the structure.

**Figure 7 insects-14-00839-f007:**
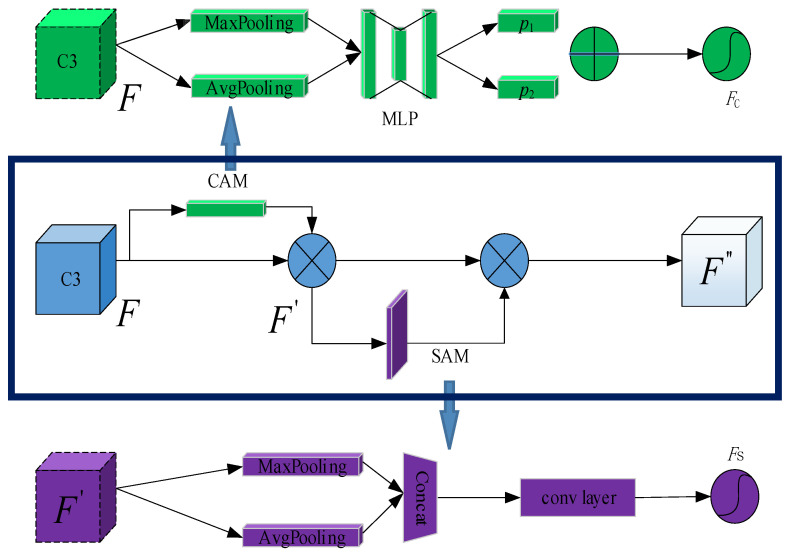
CBAM Attention Mechanism. Consisting of a Channel Attention Module (CAM) and a Spatial Attention Module (SAM).

**Figure 8 insects-14-00839-f008:**
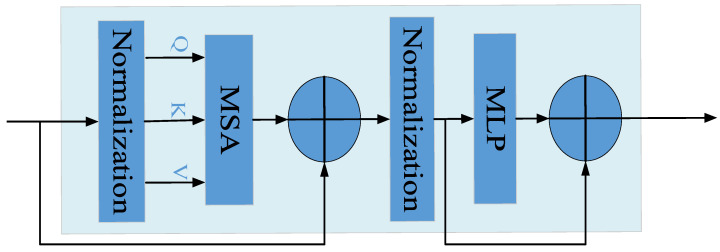
Transformer encoder. It is composed of a multi-head self-attention mechanism (MSA) and multi-layer perceptron (MLP).

**Figure 9 insects-14-00839-f009:**
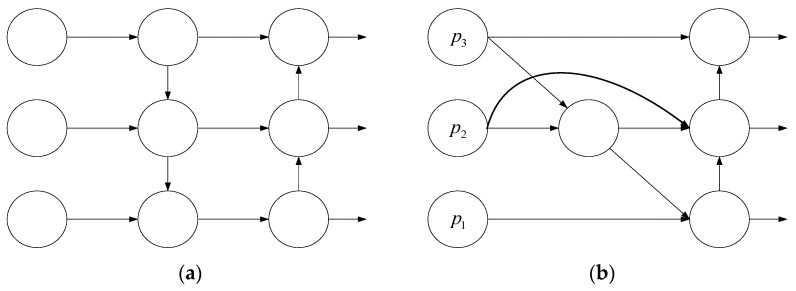
PANet and BiFPN Network structure diagrams. (**a**) PANet and (**b**) BiFPN.

**Figure 10 insects-14-00839-f010:**
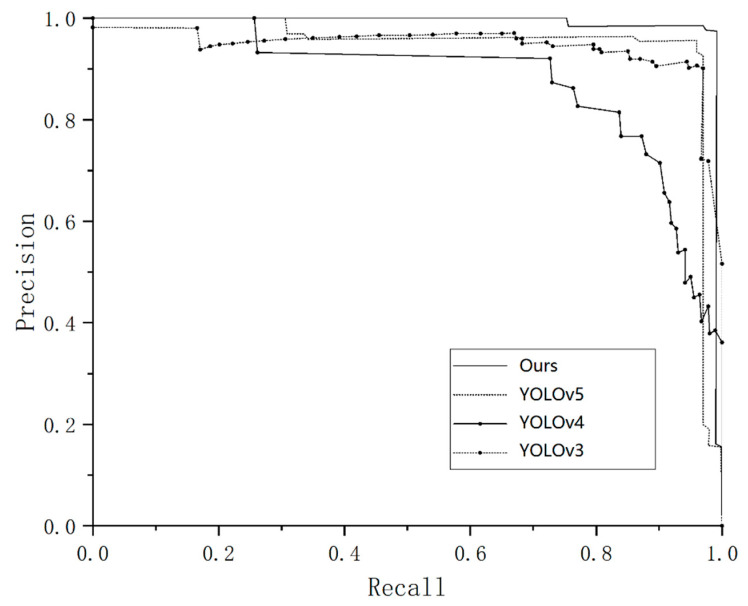
P–R curves of YOLOv3, YOLOv4, YOLOv5, and our improved YOLOv5, where the *x*-axis represents Recall, and the *y*-axis represents Precision.

**Figure 11 insects-14-00839-f011:**
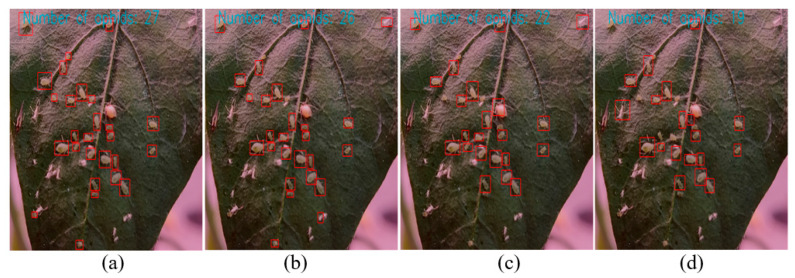
Comparison of recognition effects of different models on aphids on pepper crops. (**a**) Improved model recognition results; (**b**) YOLOV5 recognition results; (**c**) YOLOv4 recognition results; and (**d**) YOLOv3 recognition results.

**Figure 12 insects-14-00839-f012:**
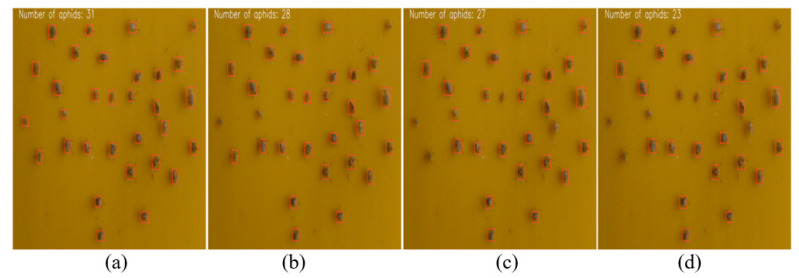
Comparison of aphid recognition effects of different models on yellow trap boards. (**a**) Improved model recognition results; (**b**) YOLOV5 recognition results; (**c**) YOLOv4 recognition result; and (**d**) YOLOv3 recognition result.

**Figure 13 insects-14-00839-f013:**
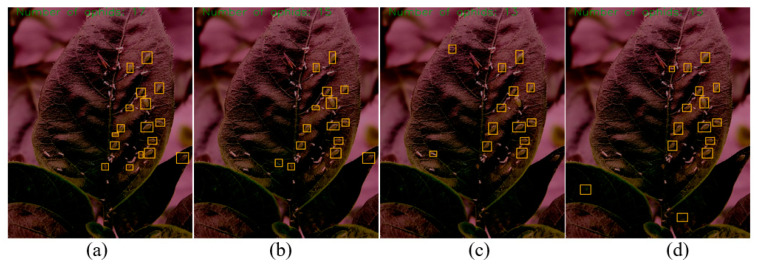
Comparison of aphid recognition effects between different models under low light intensity. (**a**) Improved model recognition results; (**b**) YOLOV5 recognition results; (**c**) YOLOv4 recognition result; and (**d**) YOLOv3 recognition result.

**Table 1 insects-14-00839-t001:** List of main abbreviations.

Abbreviations	Full Name
YOLO	You only look once
CNN	Convolutional neural network
Mask R-CNN	Mask region with the convolutional neural network
SSD	Single shot multi-box detector
RCNN	Region with the convolutional neural network
CBAM	Convolutional block attention mechanism
CAM	Channel attention module
SAM	Spatial attention module
MLP	Multilayer perceptron
MSA	Multi-head self-attention mechanism
BiFPN	Bi-directional feature pyramid network
PANet	Path aggregation network
SiLU	Sigmoid-weighted linear units
*mAP*	Mean average precision
*AP*	Average precision
*P*	Precision
*R*	Recall
*IOU*	Intersection over union

**Table 2 insects-14-00839-t002:** Comparison of experimental results for YOLOV3, YOLOv4, YOLOv5, and proposed model in terms of different performance indicators including *P*, *R*, *mAP*, and inference time.

Model	*P*/%	*R*/%	*mAP*@0.5/%	Inference Time/ms
Ours	0.991	0.991	0.993	9.4
YOLOv5	0.857	0.895	0.873	7.5
YOLOv4	0.829	0.872	0.829	17.3
YOLOv3	0.815	0.833	0.762	12.9

**Table 3 insects-14-00839-t003:** Ablation test results for different improvement points. It includes CBAM, BiFPN, and Transformer.

Model	CBAM	BiFPN	Transformer	*P*/%	*R*/%	*mAP*@0.5/%
Ours	✓	✓	✓	0.991	0.991	0.993
YOLOv5				0.857	0.895	0.873
YOLOv5-A	✓			0.937	0.925	0.912
YOLOv5-B		✓		0.880	0.933	0.902
YOLOv5-C			✓	0.926	0.951	0.922

## Data Availability

The datasets used and/or analyzed during the current study are available from the corresponding author on reasonable request.

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
