# Peer review of "Aphid Recognition and Counting Based on an Improved YOLOv5 Algorithm in a Climate Chamber Environment"

_insects, 2023, doi:10.3390/insects14110839_

Round 1

Reviewer 1 Report

Comments and Suggestions for Authors

In their manuscript, Li et al. investigate how fine-tuning the YOLO object detection algorithm can increase the efficiency of automatized recognition of aphids in climate chambers. The manuscript is very technical and the authors did not help the readers to understand the message. Although I am not completely new to YOLO and object detection, I struggled to understand what and why the authors did. This, at least partially, stems from the rather poor English but also from the lack of logical flow and a structure consistent with IMRD sections normally used in scientific papers.

I have several major issues with this manuscript and I, unfortunately, believe it is not yet ready for publication. In order to help the authors when they revise the manuscript, I added some crucial weaknesses below which, after addressed, will increase the chance of acceptance in their next submission.

·       Although the target organisms are insects, the manuscript is way too technical and focused on a very narrow segment of informatics to fit into an entomology-oriented journal. Basically, this manuscript could talk about rats, pigeons, or candle holders, but would not change highly the text because the focus is on the algorithm and not on the organism itself. Even the authors did not put any emphasis on the aphid target – we do not even know which species we are dealing with.

·       The Introduction starts by drawing attention to a problem, which would be absolutely fine if:

o   The authors put the problem into a global perspective and would bring up not only separated experiments in China among their references.

o   The basic terminology, at least the one needed for understanding the methods, should be explained and more focus should be added to the technical difficulties. Also, ALL abbreviations should be explained at their first use.

o   Since the manuscript deals with fine-tuning the YOLO algorithm, the original YOLO (up to version 3) and the used version should be at least cited.

o   Also, I cannot understand how “artificial climate chambers for cultivation” can ease vegetable demands, since these (according to my best knowledge) are more used in experimental settings rather than production. The authors do not provide relevant references, and thus this claim cannot be checked.

o   The authors claim “To our knowledge, there have been not reports of aphid recognition by using the YOLO algorithm yet.” but this claim shows how little the authors investigated the topic. Just a very rapid Google search on the keywords “aphids” and “YOLO” brought up for me at least 10 relevant hits.

·       The Materials and Methods are definitely not sufficiently explained for either replicating the study or understanding the methodology used.

o   Most of the abbreviations used are neither explained nor referenced properly. Even Labelimg, the annotation tool is not cited.

o   No details are available on how and with what camera (let alone the settings) were taken. Personally, I cannot understand why two different image darkness settings had to be used because cameras are absolutely capable of optimising the amount of light reaching their sensor. If this was technically not possible, why artificial lighting was not used?

o   “Since it is difficult to obtain a mass of data” – I doubt this is true. Aphids reproduce particularly fast and large populations can be reared within short periods of time in experimental settings.

o   1000 sheets AFTER augmentation does not seem to be enough, yet the original number of photos taken is not mentioned. It is also not clear whether the teaching process was done separately for each batch (leaves and traps) or everything pooled.

o   From section 2.2 very few entomologists would understand this, even if this section was clearly phrased. I suggest finding a more appropriate journal and explaining the bits clearly.

·       Results does not seem to be very well structured

o   3.1 and 3.2 still belong to Material and Methods.

§  Is it Python 1.9.0. (line 289) or Pyhton 3.8.10 (line 290).

o   Although “significant” differences are claimed throughout the text this has actually been tested. Although it is not clear from the Methods, it seems that the authors ran their process only once which does not allow any comparison.

·       Although the last section is titled “Conclusions and Future Works”, there are virtually no future perspectives listed. This section suggests a high generalisability of the method which, based on this small dataset and comparison is a large exaggeration of the importance of the results.

Comments on the Quality of English Language

English is rather poor throughout the text with several grammatical errors, misused words, and unclear sentences. Full sections need to be re-written and clarified. 

Author Response

Dear reviewer,

it is always great to hear from you and get your precious and instructive corrections and guidance! We have already answered each of these comments and modified the corresponding errors in our paper. Thanks for all of these again! We sincerely look forward to your further advice if there are other omissions.

In their manuscript, Li et al. investigate how fine-tuning the YOLO object detection algorithm can increase the efficiency of automatized recognition of aphids in climate chambers. The manuscript is very technical and the authors did not help the readers to understand the message. Although I am not completely new to YOLO and object detection, I struggled to understand what and why the authors did. This, at least partially, stems from the rather poor English but also from the lack of logical flow and a structure consistent with IMRD sections normally used in scientific papers.

I have several major issues with this manuscript and I, unfortunately, believe it is not yet ready for publication. In order to help the authors when they revise the manuscript, I added some crucial weaknesses below which, after addressed, will increase the chance of acceptance in their next submission.

Response: Thanks for the reviewer’s comment. To improve the readability of the manuscript and engage a broader audience, we have reorganized the structure of the manuscript and revised the manuscript by consulting a native English speaker.

Although the target organisms are insects, the manuscript is way too technical and focused on a very narrow segment of informatics to fit into an entomology-oriented journal. Basically, this manuscript could talk about rats, pigeons, or candle holders, but would not change highly the text because the focus is on the algorithm and not on the organism itself. Even the authors did not put any emphasis on the aphid target – we do not even know which species we are dealing with.

Response: Thank you for the reviewer’s suggestion. We have more focused on the research target and clearly explained the aphid recognition and counting based on the algorithm in the revised version. The green peach aphid [Myzus persicae Sulzer (Hemiptera: Aphididae)] is one of the most threatening insect pests in pepper cultivations, which is selected as our target organism. The conventional approach based on visual inspection and human experience for recognizing pepper leaves’ insect pests is subjective and time-consuming and costly. In addition, due to large intra-class similarity and small inter-class variance in symptom phenotypes, it is difficult to recognize and count the aphid. Therefore, there is a pressing need to develop a precise, fast, and convenient approach for recognizing and counting pepper leaf pests. It is worth noting that YOLO series have been successfully used in different domains including crop pests. Therefore, an improved YOLOv5 aphid recognition algorithm is proposed for aphid recognition and counting in the paper.

The Introduction starts by drawing attention to a problem, which would be absolutely fine if:

The authors put the problem into a global perspective and would bring up not only separated experiments in China among their references.

Response: Thanks for the reviewer’s comment. We have added some references provided by researchers from other countries in the revised version.

The basic terminology, at least the one needed for understanding the methods, should be explained and more focus should be added to the technical difficulties. Also, ALL abbreviations should be explained at their first use.

Response: Thanks for the reviewer’s comment. We have explained the basic terminology and added list of main abbreviations (see Table 1) in the revised version.

Since the manuscript deals with fine-tuning the YOLO algorithm, the original YOLO (up to version 3) and the used version should be at least cited.

Response: Thank you for the reviewer’s suggestion. We have cited the original YOLO and the used version in the revised manuscript.

Also, I cannot understand how “artificial climate chambers for cultivation” can ease vegetable demands, since these (according to my best knowledge) are more used in experimental settings rather than production. The authors do not provide relevant references, and thus this claim cannot be checked.

Response: Thanks for the reviewer’s comment. We realized that the expression of the sentence may be unreasonable. Yes, artificial climate chambers are more used in experimental settings rather than production. Here the use of artificial climate chambers is for investigating pepper pest control. We have corrected the sentence in the final version.

The authors claim “To our knowledge, there have been not reports of aphid recognition by using the YOLO algorithm yet.” but this claim shows how little the authors investigated the topic. Just a very rapid Google search on the keywords “aphids” and “YOLO” brought up for me at least 10 relevant hits.

Response: Thanks for the reviewer’s comment. We have realized that the description was not properly expressed. We have corrected the expression and added the relevant literature in the revised manuscript.

The Materials and Methods are definitely not sufficiently explained for either replicating the study or understanding the methodology used.

Response: Thank you for the reviewer’s comment. We have reorganized and explained the materials and methods.

Most of the abbreviations used are neither explained nor referenced properly. Even Labelimg, the annotation tool is not cited.

Response: Thank you for the reviewer’s suggestion. We have explained and added list of main abbreviations in Table 1. In addition, the annotation tool has been cited.

No details are available on how and with what camera (let alone the settings) were taken. Personally, I cannot understand why two different image darkness settings had to be used because cameras are absolutely capable of optimising the amount of light reaching their sensor. If this was technically not possible, why artificial lighting was not used?

Response: Thanks for the reviewer’s comment. In our experiment, we have used an RGB camera with high resolution provided by China Daheng (Group) Co. LTD. In our paper, we have used different image darkness settings to investigate aphid image features at different growth stages.

“Since it is difficult to obtain a mass of data” – I doubt this is true. Aphids reproduce particularly fast and large populations can be reared within short periods of time in experimental settings.

Response: Thanks for the reviewer’s comment. To avoid misleading readers, we have rewritten the sentence as “When the data sample is limited for the aphid dataset”.

1000 sheets AFTER augmentation does not seem to be enough, yet the original number of photos taken is not mentioned. It is also not clear whether the teaching process was done separately for each batch (leaves and traps) or everything pooled.

Response: Thanks for the reviewer’s comment. It was a mistake and we corrected it in the revised version. Here, 1000 images were the original pepper leaf aphid images. Through the data enhancement operation, 500 augmentation images of pepper leaf aphid are obtained. So a total of 1500 images were used in the experiment. In addition, the teaching process was done separately for each batch.

From section 2.2 very few entomologists would understand this, even if this section was clearly phrased. I suggest finding a more appropriate journal and explaining the bits clearly.

Response: Thanks for the reviewer’s comment. By searching in journal insects, just a very rapid search on the keywords “deep learning” or “machine learning” or “CNN” brought up for us at least 10 relevant hits. We have observed that there have been similar theories, models and methods for insect research. Hence, we think that it is Ok for submitting it to the journal. Also, it fits the topic of the special issue.

Results does not seem to be very well structured. 3.1 and 3.2 still belong to Material and Methods.

Response: Thanks for the reviewer’s comment. We have restructured the experimental results and moved section 3.1 and 3.2 to materials and methods.

Is it Python 1.9.0. (line 289) or Pyhton 3.8.10 (line 290).

Response: Thanks for the reviewer’s comment. It was a mistake. Python 1.9.0. has been replaced with Pytorch 1.9.0 in the revised version.

Although “significant” differences are claimed throughout the text this has actually been tested. Although it is not clear from the Methods, it seems that the authors ran their process only once which does not allow any comparison.

Response: Thanks for the reviewer’s comment. To verify the feasibility and efficiency of the proposed method, several experiments were carried out. We ran the experimental process multiple times. The performance indicators’ results were obtained by averaging 5 runs. Also, ablation experimental results were the same.

Although the last section is titled “Conclusions and Future Works”, there are virtually no future perspectives listed. This section suggests a high generalisability of the method which, based on this small dataset and comparison is a large exaggeration of the importance of the results.

Response: Thanks for the reviewer’s comment. We have revised the conclusion part. We have summarized the advantages of the proposed algorithm and explained what steps remain to apply the method in the real world.

Reviewer 2 Report

Comments and Suggestions for Authors

The work is interesting, timely and very well presented.

Expressions like "Moreover, in complex environments, the robustness of the model cannot be guaranteed.", "Due to the randomness of aphid migration, simply using insect traps to achieve pest counting cannot accurately reflect the internal pest problems in the environment.", "The structure of the Transformer encoder decoder has achieved good results in natural processing languages.", "It improves the overall performance of the model in different recognition scenarios."  require either significantly more details or citations to academic peer-reviewed works in order to stand.

It is not clear if the CBAM and BiFpn are part of the key contributions of the work or are merely preexisting methods that are used herein - please make the expression more evident.

Although not necessary to be implemented / addressed, but for reasons of completeness, it would also be interesting to show the effect of YOLOv5 size to the results of the proposed method.

The mention of 30k, 15k etc lux in photo taking processes, given the expected dimensions of the plant enclosures, seems a bit high, but not necessarily wrong. Please verify, once again, the correct values.

The hyperparameters of the network require at least an intuitive explanation as to their selected values or at best a short sensitivity analysis showing their effect to the received results.

Comments on the Quality of English Language

Minor English language proofing would be beneficial to the work (e.g. "It requires a complex insect pests image feature design." requires restructuring, "alternated alternation" and "detector to detect" would be better off altered, "enhancement method, expand the dataset" -> "enhancement method, the dataset was expanded" -> "1000 sheets" is not clear what it means, lines 163-164 are in a form not suited to the expression of papers and thus require restructuring, explain what does CBAM, BiFpn stand for, "), After" -> "). After", "respectively. represents" -> "respectively. Represents", "mAP@0.5 )As an" -> "mAP@0.5 ) as an", "rates, mAP@0.5 When" -> "rates, mAP@0.5. When", "to 0.5, The" -> "to 0.5, the" and many more throughout the work)

Author Response

Dear reviewer,

it is always great to hear from you and get your precious and instructive corrections and guidance! We have already answered each of these comments and modified the corresponding errors in our paper. Thanks for all of these again! We sincerely look forward to your further advice if there are other omissions.

The work is interesting, timely and very well presented.

Expressions like "Moreover, in complex environments, the robustness of the model cannot be guaranteed.", "Due to the randomness of aphid migration, simply using insect traps to achieve pest counting cannot accurately reflect the internal pest problems in the environment.", "The structure of the Transformer encoder decoder has achieved good results in natural processing languages.", "It improves the overall performance of the model in different recognition scenarios."  require either significantly more details or citations to academic peer-reviewed works in order to stand.

Response: Thank you for the reviewer’s suggestion. We have revised these expressions in the final manuscript. We have provided more detail to describe for some expressions, like "Moreover, in complex environments, the robustness of the model cannot be guaranteed." It was rewritten as “Moreover, there are still limitations such as robustness performance, computing efficiency, inaccuracy, and lack of generalization capability in complex environments [9, 10]”. Other expressions have been explained by adding citations, which could be seem in the revised version.

It is not clear if the CBAM and BiFpn are part of the key contributions of the work or are merely preexisting methods that are used herein - please make the expression more evident.

Response: Thanks for the reviewer’s comment. CBAM and BiFPN are part of the key contributions. Since the attention mechanism can be used to analyze complex scene information more quickly and efficiently. CBAM is introduced in the YOLOv5 backbone layer. It can enhance the model's feature extraction ability for targets, making it more focused on aphid target information. When BiFPN structure is employed in the architecture of YOLOv5, Multi-scales features of the green peach aphid can be fused and more spatial and contextual information can be retained to improve the overall detection accuracy of the model in different aphid detection scenarios.

Although not necessary to be implemented / addressed, but for reasons of completeness, it would also be interesting to show the effect of YOLOv5 size to the results of the proposed method.

Response: Thanks for the reviewer’s suggestion. In the manuscript, the proposed YOLOv5 model size was 42.6MB. In our previous article (Dai Min, DORJOY M M H, Miao Hong, Zhang Shanwen. A New Pest Detection Method Based on Improved YOLOv5m [J]. Insects, 14(1): 54.), we investigated the effect of different YOLO series’ model sizes to the results. So we have not considered the effect of YOLOv5 size to the results in the manuscript.

The mention of 30k, 15k etc lux in photo taking processes, given the expected dimensions of the plant enclosures, seems a bit high, but not necessarily wrong. Please verify, once again, the correct values.

Response: Thanks for the reviewer’s comment. We have checked that they were the correct values. As we know it, the light saturation point of pepper is 30000lx. So we have considered the special situations.

The hyperparameters of the network require at least an intuitive explanation as to their selected values or at best a short sensitivity analysis showing their effect to the received results.

Response: Thanks for the reviewer’s comment. YOLOv5 has many hyper-parameters for training settings. To facilitate comparison and analysis, the default values of hyper-parameters are considered for YOLO series. Some hyper-parameters such as learning rate, momentum coefficient, weight-decay coefficient, and iteration times should be taken into account comprehensively. Learning rate controls a model parameters’ update speed. If learning rate is too excessive, it can easily make model non-convergent, and if it is too small, the training speed will be slower. In general, the range of learning rate is 0.001-0.01. To improve the training speed, the initial learning rate is set to 0.01. Momentum and weight-decay coefficients are two important parameters in the optimizer. The former controls the stability of optimization direction, and the larger its value is, the more stable the optimization direction is; the latter controls the size of L2 regularization, and the smaller its value is, the less the contribution of the regulation to the model loss is. Thus we selected the default values of momentum (i.e., 0.937) and weight-decay (i.e., 0.0005). To avoid overfitting, we reduced epochs to 200 iteration times.

Minor English language proofing would be beneficial to the work (e.g. "It requires a complex insect pests image feature design." requires restructuring, "alternated alternation" and "detector to detect" would be better off altered, "enhancement method, expand the dataset" -> "enhancement method, the dataset was expanded" -> "1000 sheets" is not clear what it means, lines 163-164 are in a form not suited to the expression of papers and thus require restructuring, explain what does CBAM, BiFpn stand for, "), After" -> "). After", "respectively. represents" -> "respectively. Represents", "mAP@0.5 )As an" -> "mAP@0.5 ) as an", "rates, mAP@0.5 When" -> "rates, mAP@0.5. When", "to 0.5, The" -> "to 0.5, the" and many more throughout the work)

Response: Thank you for the reviewer’s suggestion and correction. The corresponding sentences have been restructured and corrected in the revised version.

Reviewer 3 Report

Comments and Suggestions for Authors

The MS Aphid Recognition and Counting Based on Improved YOLOv5 2 in Climate Chamber Environment presents a method for counting aphids using image analysis. It is important study and can answer an increasing need for methods to automatically assess insect numbers. A key problem with the MS it seems more written for readers of image analysis journals than those who would normally read Insects. The article would get more readers if it tried to engage a broader audience.

TITLE - gives a reasonable indication of the content.

KEYWORDS - OK, though it might be sensible to replace "YOLOv5 " as it occurs in the title, so would not broaden the search

NOMENCLATURE: Readers of Insects would normally expect Aphids to be given their Latin binomial and authority and date at the first mention or at the very least a general scientific name (e.g. Superfamily: Aphidoidea Geoffroy, 1762) if the species is not known. 

INTRODUCTION: I think it would be useful to reference other methods of insect counting beyond image analysis. Insect detection can work both in terms of the detection of the insects or the actual damage it causes, Some use the metabolic CO2 [Koestler et al, 2000]. Passive acoustic methods for example can listen to the sound produced by the insect [Hickling et al., 1997]. Microwaves and radar have also been used to detect the presence of termites. Ultra-wideband radio waves in the microwave region penetrate into materials and detect active pest infestations [Herrmann et al., 2013]. X-ray computed tomography has been used to detect dry wood termite infestations [Arbat et al., 2021]. Trapping and detection with automated pheromone attractant lures with the battery‐powered SightTrap™ system and its accompanying software ForesightIPM™ has been used to detect the presence of Tineola bisselliella. I do not know what references are best in agriculture, but some examples from areas I know better are: e.g. (1) Arbat, S., Forschler, B. T., Mondi, A. M., & Sharma, A. (2021). The case history of an insect infestation revealed using x-ray computed tomography and implications for museum collections management decisions. Heritage, 4(3), 1016-1025. (2) Herrmann, R., Sachs, J., Fritsch, H. C., & Landsberger, B. (2013). Use of ultra-wideband (UWB) technology for the detection of active pest infestation. In Conference in Vienna, Austria 2013 (p. 68). (3) Hickling, R., Wei, W., & Hagstrum, D. W. (1997). Studies of sound transmission in various types of stored grain for acoustic detection of insects. Applied Acoustics, 50(4), 263-278. (4) Koestler, R. J., Sardjono, S., & Koestler, D. L. (2000). Detection of insect infestation in museum objects by carbon dioxide measurement using FTIR. International biodeterioration & biodegradation, 46(4), 285-292.

Although reliance on the human eye for judgment seems tedious it can be useful for detecting other species of aphids, which might readily be missed in image analysis.  It may be worth commenting on the error of detecting other similar insects, but mis-counting them as aphids.

I have used only the simplest forms of image analysis, so am not at all familiar with YOLOv5; thus I am not able to make expert comment. However I worried about the assessment of Table 1. What method was used to determine the actual number of aphids in order to determine different performance indicators. 

I understand it is necessary to expand the data set to avoid the overfitting of the model, but do the methods of enhancing the dataset introduce errors.  It seemed for me there was the potential for this to be easier for the software than presenting multiple new images. 

It seems particularly important for real-word studies to be able recognise and count aphids on leaves rather than traps. I would have value a sense of how much performance might be degraded on leaves compared with traps. 

Further there seems to be an even more difficult problem because early in the paper we learn that aphids are found back of leaves, or curled up in young new leaves, hidden in various positions. How does the proposed method help with this problem?

CONCLUSION

The conclusion seems very brief and would benefit from explaining what steps remain to apply the method in the real world. 

MINOR POINTS

Fig. 1 seems to be 3 figures that I would prefer labelled (a) (b) (c)

Keep variables in italic script. I think the variable should not be replaced with words as in Eq. 10 and 11. 

Comments on the Quality of English Language

In general the MS could be understood, but it would benefit from editing as there are poor word choices in places

"directly eroding plants while carrying"  perhaps "v" is better something like "consuming"

of vegetable plants  - possibly just on vegetables 

I did not understand "The aphids alternated alternation of generations and accumulated seriously, making"

Careful with plurals. 2.2. Methodologies  - Methodology is an uncountable noun when used in this sense so the plural is is  methods! However "methodology"  covers a plurality of things also. 

4. Conclusions and Future Works:  use work (an uncountable nous again) as the plural works means a factory

Author Response

Dear reviewer,

it is always great to hear from you and get your precious and instructive corrections and guidance! We have already answered each of these comments and modified the corresponding errors in our paper. Thanks for all of these again! We sincerely look forward to your further advice if there are other omissions.

The MS Aphid Recognition and Counting Based on Improved YOLOv5 2 in Climate Chamber Environment presents a method for counting aphids using image analysis. It is important study and can answer an increasing need for methods to automatically assess insect numbers. A key problem with the MS it seems more written for readers of image analysis journals than those who would normally read Insects. The article would get more readers if it tried to engage a broader audience.

TITLE - gives a reasonable indication of the content.

Response: Thanks a lot. We have hoped that the journal Insects would get more readers from all sides. The image analysis and processing technology has been successfully applied to many fields. It is promising to integrate the technology with the insect pests research.

KEYWORDS - OK, though it might be sensible to replace "YOLOv5 " as it occurs in the title, so would not broaden the search

Response: Thanks a lot.

NOMENCLATURE: Readers of Insects would normally expect Aphids to be given their Latin binomial and authority and date at the first mention or at the very least a general scientific name (e.g. Superfamily: Aphidoidea Geoffroy, 1762) if the species is not known.

Response: Thank you for the reviewer’s suggestion. The green peach aphid [Myzus persicae Sulzer (Hemiptera: Aphididae)] is selected as our target organism.

 INTRODUCTION: I think it would be useful to reference other methods of insect counting beyond image analysis. Insect detection can work both in terms of the detection of the insects or the actual damage it causes, Some use the metabolic CO2 [Koestler et al, 2000]. Passive acoustic methods for example can listen to the sound produced by the insect [Hickling et al., 1997]. Microwaves and radar have also been used to detect the presence of termites. Ultra-wideband radio waves in the microwave region penetrate into materials and detect active pest infestations [Herrmann et al., 2013]. X-ray computed tomography has been used to detect dry wood termite infestations [Arbat et al., 2021]. Trapping and detection with automated pheromone attractant lures with the battery‐powered SightTrap™ system and its accompanying software ForesightIPM™ has been used to detect the presence of Tineola bisselliella. I do not know what references are best in agriculture, but some examples from areas I know better are: e.g. (1) Arbat, S., Forschler, B. T., Mondi, A. M., & Sharma, A. (2021). The case history of an insect infestation revealed using x-ray computed tomography and implications for museum collections management decisions. Heritage, 4(3), 1016-1025. (2) Herrmann, R., Sachs, J., Fritsch, H. C., & Landsberger, B. (2013). Use of ultra-wideband (UWB) technology for the detection of active pest infestation. In Conference in Vienna, Austria 2013 (p. 68). (3) Hickling, R., Wei, W., & Hagstrum, D. W. (1997). Studies of sound transmission in various types of stored grain for acoustic detection of insects. Applied Acoustics, 50(4), 263-278. (4) Koestler, R. J., Sardjono, S., & Koestler, D. L. (2000). Detection of insect infestation in museum objects by carbon dioxide measurement using FTIR. International biodeterioration & biodegradation, 46(4), 285-292.

Response: Thanks a lot. According to the reviewer’s suggestion, we have added the useful references to the introduction part.

Although reliance on the human eye for judgment seems tedious it can be useful for detecting other species of aphids, which might readily be missed in image analysis.  It may be worth commenting on the error of detecting other similar insects, but mis-counting them as aphids.

Response: Thanks for the reviewer’s comment. It is also challenging to accurately identify different insect pests with similar symptoms. We are trying to break through the bottleneck problem by using the integration of hyperspectral, thermal infrared, and RGB techniques with artificial intelligence in the future work.

I have used only the simplest forms of image analysis, so am not at all familiar with YOLOv5; thus I am not able to make expert comment. However I worried about the assessment of Table 1. What method was used to determine the actual number of aphids in order to determine different performance indicators.

Response: Thanks for the reviewer’s comment. Different performance indicators like P, R, mAP, and inference time in the table have been presented to verify the aphid detection accuracy of the proposed model. Our YOLO method enhanced the model's feature extraction ability for aphid small targets under different light sensitivity and aggregation levels. When the feature information of each aphid was recognized accurately, automatic counting of aphids can be achieved.

I understand it is necessary to expand the data set to avoid the overfitting of the model, but do the methods of enhancing the dataset introduce errors. It seemed for me there was the potential for this to be easier for the software than presenting multiple new images.

Response: Thanks for the reviewer’s comment. Data enhancement makes limited data produce value equivalent to more data without substantially increasing it. The operations of data enhancement may introduce training samples with different labels. If the pixel points outside the image are preserved for different training samples, additional errors (like training error) may be introduced into the training data. Although it is better to collect as many "natural" training samples as possible, data enhancement can be used to overcome the limitations of small data sets when it is not possible to increase real training samples.

It seems particularly important for real-word studies to be able recognise and count aphids on leaves rather than traps. I would have value a sense of how much performance might be degraded on leaves compared with traps.

Response: Thanks for the reviewer’s comment. By comparing identification and counting of aphids on the trap board and pepper leave in Figure 11and Figure 12, we can obtain that the number of aphids is 27 on leaves while the number of aphids is 31 on traps. Although the performance of the proposed method reduced by 14.8% for real-world scenarios, there is a 22.7%-42.1% improvement over YOLOv3 and YOLOv4.

Further there seems to be an even more difficult problem because early in the paper we learn that aphids are found back of leaves, or curled up in young new leaves, hidden in various positions. How does the proposed method help with this problem?

Response: Thanks for the reviewer’s comment. To deal with the more difficult problem, several aspects could be taken into account in the future work. First, the multi-lens and multi-angle camera will be employed to expand the scope of visual images; second, a multi-sensor collaboration method will be used to obtain more information features, by identifying the number of aphids and associating them with other environmental factors, a multi-source information fusion model of pest environmental factor disease can be established to monitor the development trend of insect pests.; third, collecting large-scale datasets, the missing information hidden in various positions will be identified by using our proposed YOLOv5 network architecture and the creation of a complete image will be simulated by using artificial intelligence.

CONCLUSION

The conclusion seems very brief and would benefit from explaining what steps remain to apply the method in the real world.

Response: Thank you for the reviewer’s suggestion. To make the method practically applicable to farmers, an application (APP) capable of deploying the proposed model on various types of mobile devices is needed. Then, based on the results of green peach aphid recognition and counting, decision-making strategies are generated to guide pesticide spraying.

MINOR POINTS

Fig. 1 seems to be 3 figures that I would prefer labelled (a) (b) (c)

Keep variables in italic script. I think the variable should not be replaced with words as in Eq. 10 and 11.

Response: Thank you for the reviewer’s suggestion. We have revised these minor points.

In general the MS could be understood, but it would benefit from editing as there are poor word choices in places

"directly eroding plants while carrying"  perhaps "v" is better something like "consuming"

of vegetable plants  - possibly just on vegetables

Response: Thank you for the reviewer’s suggestion. We have corrected poor word choices in the revised version.

I did not understand "The aphids alternated alternation of generations and accumulated seriously, making"

Response: Thanks for the reviewer’s comment. We have rewritten the sentence. i.e., “The aphid alternates between generations and accumulates seriously, which makes it difficult to count the aphids.”

Careful with plurals. 2.2. Methodologies  - Methodology is an uncountable noun when used in this sense so the plural is  methods! However "methodology"  covers a plurality of things also.

Response: Thanks for the reviewer’s comment. We have replaced Methodologies with Deep learning-based methods.

  1. Conclusions and Future Works: use work (an uncountable nous again) as the plural works means a factory

Response: Thanks for the reviewer’s comment. We have replaced Conclusions and Future Works with Conclusion.

Round 2

Reviewer 1 Report

Comments and Suggestions for Authors

The Authors made a great effort to improve the manuscript and it, indeed, reads better now than the previous version and clarifies some missing points. Despite their effort, I still cannot recommend this manuscript for publication in the Insects for four major reasons:
1) In my opinion it does not fit the Aims and scope of the journal, it is clearly a computational-focused manuscript with very little relevance to entomology. Whilst I also can see similar topics being published in Insects, most of those are less technical and more insect-focused. 
2) An entirely different issue: I still cannot see how the study was replicated and how tests were carried out to statistically compare whether the described method is really superior to others. The Authors claim in their reply "to verify the feasibility and efficiency of the proposed method, several experiments were carried out. We ran the experimental process multiple times. The performance indicators’ results were obtained by averaging 5 runs." but I could not find this information in the text and neither were the statistical tests shown. If values shown are means, providing the linked standard error/confidence limits would also be necessary, as would be providing information on how the replications were done (e.g. randomising training-test sets, changing training and test proportions etc). 
3) Discussion is still not discussing the results of the study but future perspectives. No words about limitations, in what setting this model could be used, how does this fit into the already booming ecological object detection "ecosystem", and how the results can be influenced by the ecology of the target insect or plant, or sampling method (yellow trap).
4) Structuring and clear, yet concise, wording (not English grammar per se) still needs a substantial amount of attention. 

Comments on the Quality of English Language

The English has improved but in several sections, it still does not read well, it can be slightly improved. E.g., "The aphid alternates between generations and accumulates seriously, which makes it difficult to count the aphids". Please note, that this is only an example, there are several similarly unclear sentences in the text.

Author Response

The Authors made a great effort to improve the manuscript and it, indeed, reads better now than the previous version and clarifies some missing points. Despite their effort, I still cannot recommend this manuscript for publication in the Insects for four major reasons:
1) In my opinion it does not fit the Aims and scope of the journal, it is clearly a computational-focused manuscript with very little relevance to entomology. Whilst I also can see similar topics being published in Insects, most of those are less technical and more insect-focused.

Response: Thanks for the reviewer’s comment. To fit the Aims and scope of the journal, we have focused on the application of computer vision and image processing techniques in agricultural pest management. We have improved the manuscript to help the readers to understand the text in the revised version.

2) An entirely different issue: I still cannot see how the study was replicated and how tests were carried out to statistically compare whether the described method is really superior to others. The Authors claim in their reply "to verify the feasibility and efficiency of the proposed method, several experiments were carried out. We ran the experimental process multiple times. The performance indicators’ results were obtained by averaging 5 runs." but I could not find this information in the text and neither were the statistical tests shown. If values shown are means, providing the linked standard error/confidence limits would also be necessary, as would be providing information on how the replications were done (e.g. randomising training-test sets, changing training and test proportions etc).

Response: Thanks for the reviewer’s comment. First, all tested and compared models were carried out in the same experimental settings and evaluating indicators environment. As for YOLO series, their hyperparameter setting were the same, which can be found in experimental settings section. For fair comparison, the same evaluation indicators were given, which can be found in evaluating indicators section. Next, aphid dataset was input into each model for training and testing the data. Then, the given indicators would be calculated according to the number of positive and negative samples predicted. Sorry for the previous unreasonable reply.

3) Discussion is still not discussing the results of the study but future perspectives. No words about limitations, in what setting this model could be used, how does this fit into the already booming ecological object detection "ecosystem", and how the results can be influenced by the ecology of the target insect or plant, or sampling method (yellow trap).

Response: Thank you for pointing this out. We have rewritten the discussion section as in the following:

In this study, we proposed an improved YOLOv5 approach based on deep learning to recognize and count the green peach aphid on pepper crops in the climate chamber environment. Our findings demonstrated that the improvement in the overall performance of the proposed model was achieved. Concretely speaking, by introducing CBAM, the spatial and channel dimensions of aphid features are compressed, the network model can better obtain the key information of the feature map. Thus it can effectively improve the accuracy of aphid recognition on pepper leaves. Using the feature fusion method of BiFPN structure enhances the YOLOv5 neck, further improving the computation efficiency and recognition accuracy of aphids. In addition, by adding the Transformer encoder, the extraction and expression of global features of the aphid can be achieved better. Also, it has less computational complexity. Compared to other YOLO algorithms (such as YOLOv3, YOLOv4, and YOLOv5), our method achieved recognition accuracy and recall rates of 99.1% and 99.1%, the mAP@0.5 of 99.3%, and the inference time of 9.4ms, which is superior to them.

By investigating the extent of aphid infestations on pepper crops based on the proposed model, we found that aphid outbreaks and identification environments have an impact on the improved YOLOv5 model in the climate chamber. At the early stage of aphid outbreaks, there were limited aphids on pepper crops. Thus the method can identify and count aphids on pepper leaves very well by comparing with other YOLO algorithms. At the late stage of the aphid outbreaks, we needed to add yellow trap boards to recognize and count them as the aphids propagated continuously and accumulated seriously. On the yellow trap board, the aphid was single and the background complexity was smaller than pepper crops, making it easy for the model to achieve aphid recognition and counting. In addition, the low light intensity made it difficult to count aphids on pepper crops. Fortunately, due to the introduction of CBAM, the model can still show perfect performance with regard to aphid recognition and counting.

Additionally, compared to conventional methods such as artificial pest recognition and counting, our aphid recognition and counting model provides an efficient, fast, and cost-effective method for growers. At the same time, it is convenient and feasible for farmers who may not have the expertise to recognize and count the green peach aphid on pepper crops. Finally, it can perform both offline and online detection in the field by being non-destructive.

Despite the favorable performance demonstrated in the task of pepper leaf aphid recognition and counting, there are certain limitations for further research and improvement. Further, there seems to be an even more difficult problem because early in the paper we learn that aphids were found back of leaves, or curled up in young new leaves, hidden in various positions. To deal with the more difficult problem, several aspects could be taken into account in the next work. First, the multi-lens and multi-angle camera will be employed to expand the scope of visual images; second, a multi-sensor collaboration method will be used to obtain more information features, by identifying the number of aphids and associating them with other environmental factors, a multi-source information fusion model of pest environmental factor disease can be established to monitor the development trend of insect pests.; third, collecting large-scale datasets, the missing information hidden in various positions will be identified by using our proposed YOLOv5 network architecture and the creation of a complete image will be simulated by using artificial intelligence.

4) Structuring and clear, yet concise, wording (not English grammar per se) still needs a substantial amount of attention.

Response: Thank you. We have reorganized the structure of the manuscript and accordingly modified wording in the revised version.

5)The English has improved but in several sections, it still does not read well, it can be slightly improved. E.g., "The aphid alternates between generations and accumulates seriously, which makes it difficult to count the aphids". Please note, that this is only an example, there are several similarly unclear sentences in the text.

Response: Thanks for the reviewer’s comment. To improve the readability of the manuscript, we have re-described unclear sentences in the text. For example, the sentence pointed out had been replaced with “The aphids propagate continuously and accumulate seriously, which makes it difficult to count them.” The other unclear sentences had been revised (in red).

Reviewer 3 Report

Comments and Suggestions for Authors

The authors have worked hard. My only minor concern which can be addressed in editing was the need to put Latin binomials in italic script. 

Comments on the Quality of English Language

Will be OK with minor copyediting

Author Response

The authors have worked hard. My only minor concern which can be addressed in editing was the need to put Latin binomials in italic script. 

Response: Thanks for the reviewer’s suggestion.  We have accordingly revised Latin binomials.

Will be OK with minor copyediting

Response: Thank you. Yes, we have checked the manuscript and revised it in the final version.